# Empirical data drift detection experiments on real-world medical imaging data

Ali Kore[1], Elyar Abbasi Bavil[2], Vallijah Subasri [3], Moustafa Abdalla[4], Benjamin Fine [5,6], Elham Dolatabadi[1,7] & Mohamed Abdalla [5] ✉

While it is common to monitor deployed clinical artificial intelligence (AI) models for performance degradation, it is less common for the input data to be monitored for data drift – systemic changes to input distributions. However, when real-time evaluation may not be practical (eg., labeling costs) or when gold-labels are automatically generated, we argue that tracking data drift becomes a vital addition for AI deployments. In this work, we perform empirical experiments on real-world medical imaging to evaluate three data drift detection methods' ability to detect data drift caused (a) naturally (emergence of COVID-19 in X-rays) and (b) synthetically. We find that monitoring performance alone is not a good proxy for detecting data drift and that drift-detection heavily depends on sample size and patient features. Our work discusses the need and utility of data drift detection in various scenarios and highlights gaps in knowledge for the practical application of existing methods.

As the number of artificial intelligence (AI) tools in medicine grows, patients may increasingly be evaluated by physicians who employ a wide gamut of supportive clinical machine learning algorithms. To ensure the safe use of these algorithms, researchers and practitioners are developing a wide-set of evaluative and deployment best practices[1–3]. For example, it is widely recognized that algorithms tend to underperform when applied to populations that differ from those they are trained on[4–6]. To counter this generalization gap, many researchers have increased the diversity represented in datasets used for AI development[5,7]. Others have presented possible technical solutions of generalizability, such as improved data preprocessing[8], neural network normalization algorithms[9], and an assortment of training strategies[10].

Compared to the level of attention given to improving the generalizability of machine learning models, there has been relatively little work focusing on monitoring models deployed in production for changes in the population which may lead to failure of generalizability—so called data drift. Data drift, defined as the systematic shift in the underlying distribution of input features[11], can cause models' performance to deteriorate[11–13] or behave in

unexpected ways, which can pose a threat to the safety of patients. For example, changes to the demographics of the population served over time (e.g., immigration to a city) may unknowingly change the distribution of input data to an AI model—classes which were previously not defined may now require definition. In such a situation, failing to re-train the model to also predict the novel pathologies may deleteriously impact patient safety (e.g., a chest X-ray binary classifier may now erroneously classify the novel pathology as normal). Drift detection enables healthcare providers to follow technical[14–16] and regulatory[17] best-practices guidelines for machine learning which require a model be thoroughly evaluated on the deployment population and monitored to ensure safety—detection of drifts would indicate potential risk and trigger re-evaluations. Detecting this drift enables healthcare providers to proactively intervene before risk reaches the patient and decide if the model should be revaluated, retrained, taken offline, retired or replaced. It is therefore vital that deployed algorithms are monitored for drift in the populations they serve. Not doing so risks algorithms underperformance (in the best case), or patient safety risk (in the worst case).

[1]Vector Institute, Toronto, Canada. [2]Temerity School of Medicine, University of Toronto, Toronto, Canada. [3]Peter Munk Cardiac Center, University Health Network, Toronto, ON, Canada. [4]Department of Surgery, Harvard Medical School, Massachusetts General Hospital, Boston, USA. [5]Institute for Better Health, Trillium Health Partners, Mississauga, Canada. [6]Department of Medical Imaging, University of Toronto, Toronto, Canada. [7]School of Health Policy and Management, Faculty of Health, York University, Toronto, Canada. ✉e-mail: mohamed.abdalla@mail.utoronto.ca

Previous work focusing on drift detection in clinical settings often promotes tracking changes in model performance as a proxy for data drift[11,18,19] and explaining these drifts using post hoc analysis (e.g., SHAP values (SHapley Additive exPlanations)[11,20]). While these approaches have multiple upsides in that they are often simple to implement, easy to interpret, and in certain situations, easy to act upon, they also suffer multiple weaknesses. First, the data to evaluate model performance can be difficult to obtain in a timely manner (e.g., instances where outcomes or diagnosis do not occur for days, weeks, or months later). In other situations, automated approaches to creating gold labels may not be available and it is cost-prohibitive to pay for human annotators to create diagnosis labels[21]. Even if automated approaches can be used to generate gold labels, they themselves may be affected by the data drift thus providing inaccurate labels and affecting the validity of performance monitoring (e.g., past work has shown increased negativity present in the clinical notes of obese patients[22,23], a trend which may negatively impact the accuracy of gold labels created by neural language models if there is an increase in the proportion of obese patients). Lastly, there is a growing body of statistical work contesting the use of SHAP values for feature explanations[24–26].

While more recent works also seek to directly detect data drift in clinical settings[19,27], their explorations of data drift are not systematic, often do not include examples of real-world drift, and still use decreases in performance measures to demonstrate the utility of their methods. This study seeks to address many of these issues. Exploring three methods for performing data drift detection that do not rely on ground truth labels, we study: (1) how these methods perform in the face of real-world data drift, (2) how different types of demographic drifts affect drift detection, and (3) discuss the different scenarios where drift detection approaches may not be captured by tracking aggregate model performance. Our work highlights how, in addition to using data drift detection to detect performance changes[19,27], data drift detection can also be used by practitioners to trigger re-evaluation of their models in line with best-practice clinical AI guidelines.

In this study, we perform a systematic exploration of data drift detection in AI-based chest X-ray prediction models, designed to predict diagnoses/pathologies from X-ray images, using (a) a real-world dataset with naturally occurring data drift (the emergence of COVID-19 in March 2020) and (b) synthetic drifts. We evaluate the utility of drift detection in multiple real-world scenarios to highlight when and how drift detection can be used to improve patient safety and model understanding. Furthermore, we demonstrate: (1) how drift detection can occur without a change in model performance (especially using commonly reported metrics), (2) the effect of dataset size on drift detection sensitivity, and (3) the sensitivity of the two broad approaches to drift detection (model performance-based and data-based) to different types of drift (e.g., changes in patient demographics, patient types, and pathologies).

## Results
### Overview of task, data, and approach
The task of drift detection is to ascertain whether two different sets of data (source dataset and target dataset) are from the same distribution or if the target dataset has 'drifted' from the source dataset. In this study, we define data drift to refer to the scenario where source and target samples originate from the same context but at different times; a more in-depth discussion of data drift can be found in the next section. Practically, this is typically an on-going process which is performed throughout the lifetime of an algorithm's deployment[28]. This is related to, yet separate from, the concept of generalizability of a model which can and should be used to evaluate the effect of different locations or time periods before deployment, typically only once.

In this study, we explored data drift using the task of chest radiograph disease classification. The dataset for this experiment is composed of 239,235 temporally performed chest radiographs (CXRs)

(and associated imaging reports) collected before and after the emergence of COVID-19 at Trillium Health Partners, a high-volume, full service, three-site hospital system that serves the ethnically diverse population of Mississauga, Ontario, Canada. We used a pre-trained TorchXRayVision classifier fine-tuned to predict the presence of 14 pathologies. The complete details regarding the dataset and pre-processing can be found in the Methods section.

We empirically investigated and compared the efficacy of four approaches to drift-detection: (1) tracking model performance, (2) image data-based drift detection (TorchXRay Vision AutoEncoder, henceforth: TAE), (3) model output-based drift detection (Black Box Shift Detection, henceforth: BBSD), and (4) combined image-and-output-based drift detection (henceforth: TAE + BBSD). Initially, we tested the ability of these approaches to detect real-world data drift, caused by the introduction of CXRs with COVID-19[29] during the first wave of the COVID-19 pandemic. Subsequently, to assess the robustness of these approaches, we studied if they can detect synthetic categorical drifts where we simulate changes in patient demographics and pathologies. Finally, we explored the effect of sample size on the sensitivity of drift detection approaches.

### Data-based drift detection is able to capture real-world data drift that is not captured by tracking model performance alone
Despite the common use of tracking model performance as a proxy for underlying data drift, we find that in the real-world natural experiment of data drift caused by the COVID-19 pandemic, this approach fails to capture the clinically-obvious data drift[29]. Figure 1 plots the macro-average AUROC as well as the *p*-value resulting from the TAE + BBSD combined image-and-output-based drift detection approach. We observe that the AUROC is relatively stable and does not meaningfully change in light of the first COVID-19 wave (represented by the vertical yellow line). On the other hand, TAE + BBSD detects the drift caused by the introduction of COVID-19 and the end of the first wave (though with a delay). This result seems to indicate that aggregate measures of performance are not a reliable proxy for detecting data drift (something which is expected and confirmed in later experiments). This is expected because the primary purpose of tracking model performance is not to detect drift; rather, any drift which does not affect model performance (a list of possible reasons can be found in the discussion) will not be detected by tracking model performance alone.

### The sensitivity of data drift detection depends on the feature which is synthetically enriched
Figure 2 presents the results of drift detection experiments for various synthetic drifts where we changed the underlying patient population (demographic distribution) to simulate data drift. Supplementary Fig. 2 has a similar experiment where the prevalence of pathologies was changed (instead of demographics) to simulate data drift. For each simulated data drift, we study the effect of varying amounts of data drift (from 5% increase to 50% increase). Each subplot presents the AUROC of the classification model on the right y-axis, and the results of the image data-based drift detection (TAE), model output-based (BBSD), and image-and-output-based detection (TAE + BBSD) on the left y-axis.

As observed with the COVID data drift, the AUROC is relatively stable for most of the synthetic drifts tested. This lends credence to our observation that aggregate model performance is not a reliable indicator of data drift (despite widespread use). In Supplementary Figs. 3–10, we plot other metrics (F1 score, precision, recall, and Brier score) to demonstrate the consistency of our finding. For these metrics we observe similar results: all are less sensitive than data-based drift detection approaches.

For TAE, BBSD, and TAE + BBSD, we observe that the larger the size of the synthetic drift the increased likelihood of the drift being detected. We also observe that, generally, using TAE + BBSD is usually

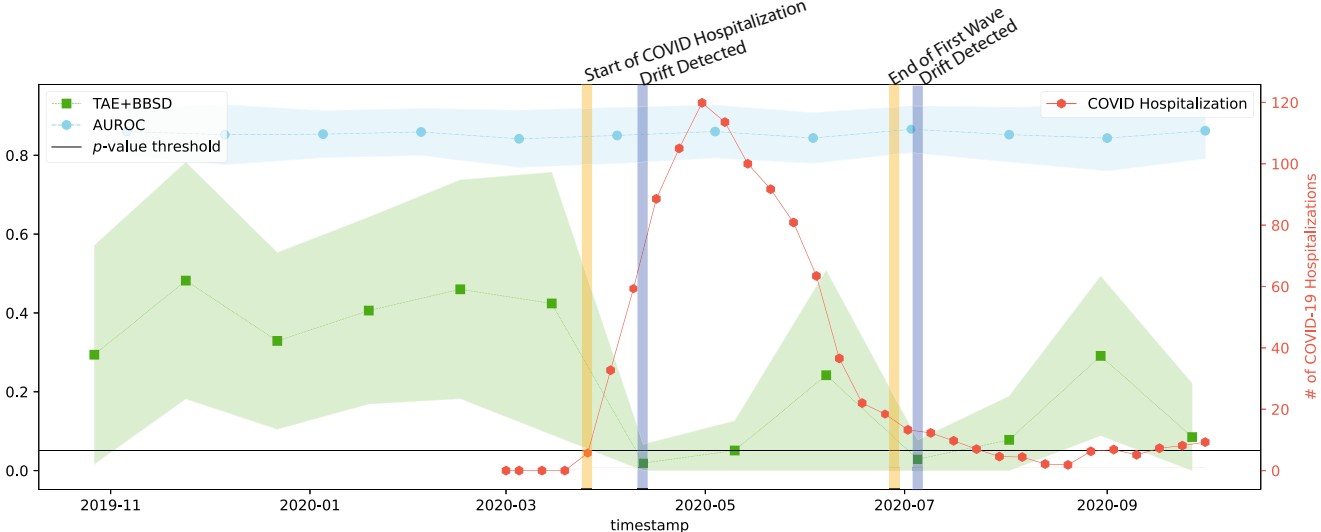

**Fig. 1 | Data drift as detected by TAE + BBSD (image-and-output-based) drift detection over time on real-world data.** The AUC is plotted in light blue and corresponds to the left y-axis. The p-value output from the TAE + BBSD approach (green) is also read against the left y-axis. p-values were calculated using a two-sided multivariate maximum mean discrepancy (MMD) statistical test with no correction for multiple comparison. The number of daily COVID hospitalizations at Trillium Health Partners is overlaid in red and can be read using the right y-axis. The horizontal line represents the value of 0.05 on the left y-axis. TAE trained autoencoder, BBSD black box shift detector, AUROC area under the curve – receiver operating characteristic.

more sensitive than just TAE and BBSD is nearly as sensitive as TAE + BBSD.

Interestingly, we can see that the sensitivity of TAE, BBSD, and TAE + BBSD varies substantially depending on the synthetically enriched feature. For example, a 5% increase in the number of patients aged 18–35 is detected by TAE + BBSD, yet it takes a 30% increase in inpatients or patients aged 65+ for the observed drift to be considered significant. It is not clear why this is the case. Initially we hypothesized that increases of rarer classes are easier to detect than increases of majority classes; patients 18–35 are only 6% of the population whereas patients 65+ are 62% of the population, Supplementary Table 1. However, this trend is reversed when looking at patient classes; inpatients are the most common patient class (57% of patients) yet are better detected than outpatients (19%). Alternatively, the level of discrimination between different groups (e.g., the variance of groups) may require a larger number of samples to detect differences[30]. More work is required to uncover the causes of these differences.

### The sensitivity of data-based drift detection is strongly correlated with sample size

In the above experiments, the source and target datasets included 4000 images each. In this section, we explore the effect of sample size on the sensitivity of TAE + BBSD. Figure 3 plots the output of TAE + BBSD across various magnitudes of data drift (increasing the proportion of male patients) for various sample dataset sizes. We observe correlation between the sample size and the sensitivity of drift detection: with a sample dataset size of 500 the p-value never drops below 0.1 and with 4000 images the p-value drops below 0.05 with a 40% increase. This finding has direct implications for the practical use of drift detection techniques: if it takes days (large radiology providers) or months (smaller providers) to perform 4000 new CXRs, then existing techniques for drift detection may not be timely enough to allow clinicians using the tool to intervene promptly.

### Aggregate performance metrics are very poor for detecting data drift

In the initial set of experiments, following from prior literature[19,31], we reported model performance using AUROC. Unfortunately, the AUROC did not substantially change in the presence of any drift (both

in the natural example of COVID-19 or in the majority of synthetic shifts). To understand why this is the case, Fig. 4, plots different performance measures: macro-average AUROC, macro-average F1 score, macro-average precision, macro-average recall, macro-average Brier score, and the F1 score breakdown for multiple pathologies for a synthetic shift increasing the proportion of patients from [Hospital Site 1] and [Hospital Site 2]. In the Supplementary Figs. 3–8, we plot the change in macro-average precision and recall for each of the tested drifts. Supplementary Figure 12 plots the change in performance measures for various degrees of data drift.

This experiment highlights three main observations. First, triple-aggregated performance measures (Fig. 4, AUROC and F1 in left subplots) which rely on the aggregation of multiple metrics (e.g., macro-average AUROC which is calculated using two other metrics: false and true positive rates, or F1-Scores which is calculated using precision and recall) across multiple classes are the least useful for drift detection. Second, individually aggregated performance measures (e.g., Fig. 4, precision, recall, and Brier Score in left subplots) are more sensitive to tested drift compared to triple-aggregated measures, but this depends on the type of drift. Lastly, the performance measures of individual classes (Fig. 4, right subplots) are the most likely to be sensitive to data drift (even doubly aggregated performance metrics).

## Discussion

Monitoring a deployed AI model is crucial in high stakes scenarios such as healthcare to ensure that any performance degradation is detected early and acted upon before patient care impact[32,33]. Current best practice includes comparing the AI model's outputs (or a sampling of the outputs) against a gold standard continuously over time[34]. The intent is to detect clinically important changes which might have been caused by changes in the underlying population of patients, disease prevalence, equipment changes, practice/referral pattern among many other factors, and facilitate intervention (retain, warn users, retire model) promptly[6,28,35].

While monitoring of model performance is a must, we argue that those using clinical AI should also use approaches which look at the input data directly to monitor for data drift for reasons related to safety, cost and reliability. This is because many types of data drift— used in this work loosely to encompass any type of drift that may occur

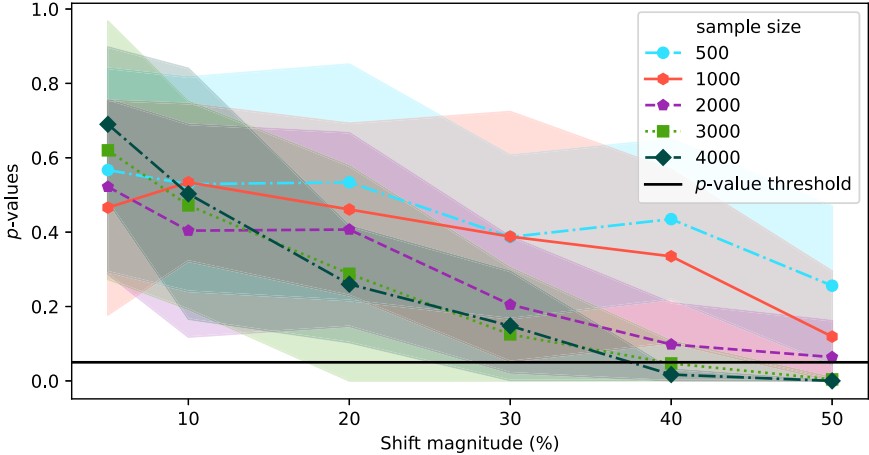

**Fig. 2 | A comparison between TAE (light blue), TAE + BBSD (red), BBSD (green), and AUROC (purple) monitoring methods for various synthetic drifts.** For each subplot, the distribution of source and target dataset is different across the following representative categories: *sex*, *institution*, *is_icu*, *patient age*, and *patient class*. The *p*-values (outputs from the TAE, TAE + BBSD, BBSD) are plotted against the left y-axis while the AUROC is plotted against the right y-axis. *p*-values were calculated using a two-sided multivariate maximum mean discrepancy (MMD) statistical test with no correction for multiple comparison. Data are presented as mean values +/− 1 standard deviation. TAE trained autoencoder, BBSD black box shift detector, AUROC area under the curve – receiver operating characteristic.

**Fig. 3 | Experiment exploring the effect of sample size on data drift detection.** Synthetic categorical shift for a target category of sex(M) with the TAE + BBSD method for a series of different sample sizes. *p*-values were calculated using a two-sided multivariate maximum mean discrepancy (MMD) statistical test with no correction for multiple comparison.

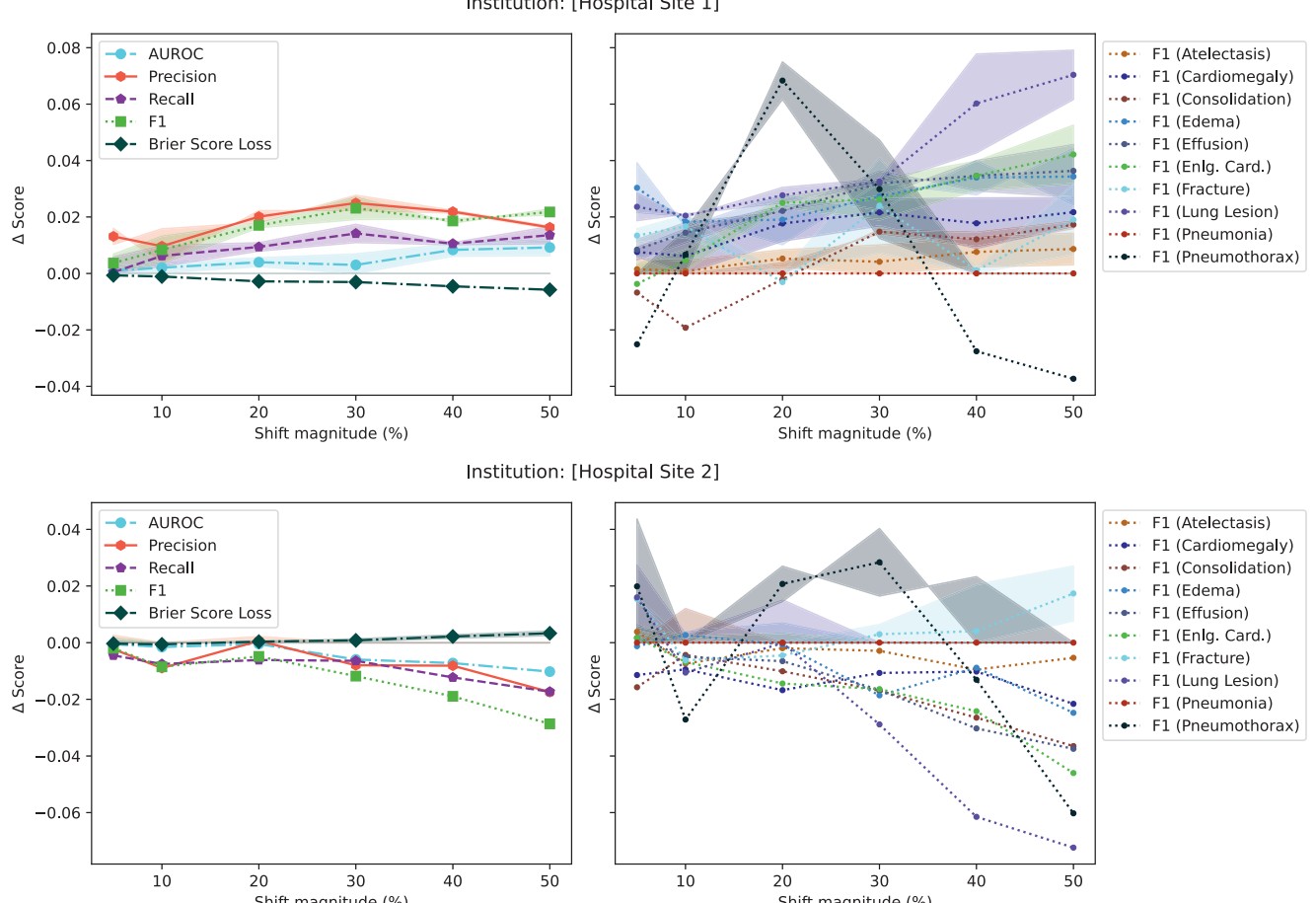

**Fig. 4 | Comparison of aggregate vs non-aggregate metrics for synthetic categorical shift for a target category of [Hospital Site 1] (4A) and [Hospital Site 2] (4B).** The plots on the left show the absolute change in score for aggregate metrics (i.e., AUROC, Precision, Recall, F1 score, and Brier Score averaged all classes). The plots on the right show the breakdown across pathologies for the F1 score. Aggregate scores change less than 3%. *p*-values were calculated using a two-sided multivariate maximum mean discrepancy (MMD) statistical test with no correction for multiple comparison. Data are presented as mean values +/− 1 standard deviation. *Eng. Card.* stands for enlarged cardiomediastinum. AUROC area under the curve – receiver operating characteristic.

(e.g., covariate shift, label shift, and concept drift)[16]—cannot always be detected by tracking performance metrics alone. For example, unless there is an extreme disparity in performance between classes, tracking performance metrics alone will not likely capture covariate shift (i.e., changes in input distribution), in contrast, we have shown, in this work, TAE is capable of doing so.

To clinically motivate the need for the approaches presented in this study, Table 1 presents multiple scenarios where data drift detection can positively impact clinical model deployment. For example, data drift detection is useful for patient safety when it is not feasible to rely on performance monitoring for timely evaluation (e.g., Scenario 1: when outcomes do not occur for days or weeks or it is cost-prohibitive to consistently produce gold-standard labels in a timely manner[21,36]). In this scenario, drift detection provides a method for tracking instances of data drift which can be used to trigger the more expensive process of data labeling for re-evaluation (i.e., only incur the cost for full evaluation when a drift is detected) to ensure patient safety. In instances where there is less time pressure (i.e., applications with longer time-frames to evaluate outcomes), data drift detection may be of less importance to ensuring patient safety, though it may be worthwhile to gain a better understanding of the patient population.

When model labels are obtained manually (Scenario 2b), performance is often evaluated using small sample sizes to minimize cost and enable timely evaluation[21]. However, if these samples are not representative of the overall patient population, there is an increased risk of

the reported model performance not being representative of the true model performance[37]. Here, data drift detection approaches can help inform those performing model evaluations when the input distributions to the model have changed, thus enabling users to update their sampling methodology to match the new patient population.

Alternatively, when performance monitoring is performed using large sample sizes (Scenario 2c), the gold labels are often generated using automated approaches (e.g., labeling CXRs by leveraging radiologist's notes and image embeddings)[31]. In these cases, data drift could affect the automated gold label generation thus making the evaluation unreliable (e.g., if there is an increase in obese patients, the increased general negativity present in the clinical notes of obese patients[22,23], could negatively impact the accuracy of gold labels created by neural language models).

Lastly, even if the gold-labels are perfect and there is no change in performance in detecting, data drift could help us understand the generalizability of the model on new populations. This monitoring for changes in demographic would be similar to how therapeutics manufactures are expected to study the demographic characteristics of patients in postmarketing surveillance of novel therapeutics[38]. When a drug is developed, for practical reasons, its approval usually depends on a trial which studies a population more limited than the population to which it is prescribed[39,40]; something similar for AI models. As such, pharmaceutical companies are being increasingly asked to actively monitor adverse events across broad demographics after approval and

**Table 1 | The utility provided by monitoring for drift detection in various scenarios faced by healthcare institutions**

| | Utility added by data drift detection |
|---|---|
| **Scenario 1: Timely gold labels are not available**.<br>*Rationale:* If timely gold labels are not available (e.g., when outcomes or diagnoses do not occur for days or weeks or where gold labels are expensive to obtain and thus not obtained on a recurring basis), data drift detection can alert users when input distributions have drifted enough to require performance re-testing. | Drift detection can inform institutions when performance needs to be measured again. |
| **Scenario 2: Timely gold labels are available**. | |
| **Scenario 2a: Performance is observed to change**.<br>*Rationale:* If timely gold labels are available, and performance is being tracked and a change in performance is noticed then data drift detection would not increase safety of deployment. | Data drift can be used to help explain causes for change in performance. |
| **Scenario 2b: Performance is not observed to change. Gold labels are produced from a sampled dataset which is manually labeled**.<br>*Rationale:* If timely gold labels are obtained through a sampling method (e.g., manual annotation of continuously curated datasets) then failure to adjust the sampling methodology to data drift can affect the validity of the evaluation. For example, consider a triaging algorithm used in a hospital on a population composed of 90% male patients and 10% female patients. The hospital is not willing to use the algorithm if the false positive alerts exceed 10%. Each month, 90 male and 10 female cases are sampled, representative of our population to calculate performance. If performance for the Male class is 90% accurate (83 correct alerts, 7 incorrect alerts) and the Female class is 80% accurate (8 correct alerts, 2 incorrect alerts), the total percentage of false positives is 9.8%. Now, consider a flip in demographics (90% female) in the population. Calculated performance in the sampled dataset would stay the same (sampled 90 males, 10 females). However, in the new population, actual performance degrades (23% false positive rate). Thus, knowing that data drift has occurred is vital to maintaining accurate representation of the sampling datasets to enable valid evaluation of model deployment. | Data Drift Detection can inform those performing model evaluations when the input distributions to the model have changed, thus enabling users to update their sampling methodology |
| **Scenario 2c: Performance is not observed to change. Gold labels are automatically captured**.<br>*Rationale:* If timely gold labels are obtained using automated methods then it may be that the gold label generation process is itself affected by data drift. As such, the observed performance cannot be trusted without additional rigor.<br>For example, there are techniques to create gold labels for ML training and evaluation which use the combination of both natural language and imaging inputs[58]. These approaches may themselves be affected by data drift thus changing the gold labels, which can ultimately hide any true change in performance. | Data Drift Detection can highlight when automated gold label creation techniques need to be re-validated. |
| **Scenario 2d: Performance is not observed to change. Gold labels are assumed to be perfect**.<br>*Rationale:* Similar to 2a. If gold labels are assumed to be perfect then drift detection is not required for safety though it can improve understanding. | Data drift can be used to help better understand the generalizability of the deployed algorithm. |

sales to patients[38], something which should also be asked of AI deployments.

Having motivated the need for drift detection as a task, in this work, we explored how automated drift detection approaches perform in the face of real-world data drifts caused by the COVID-19 pandemic as well as synthetic feature drifts. Understanding the capabilities and limitations of current approaches to drift detection can help inform deployment: whether it is safe to deploy and how often models should be retrained or evaluated, among other questions.

Our work has demonstrated that data-based approaches to drift detection can detect data drift (both natural and synthetic) in instances when monitoring model performance in aggregate cannot. While tracking non-aggregate metrics (e.g., precision and recall) for individual classes increases the sensitivity of performance-based drift detection approaches, it is cognitively tasking and very noisy (Supplementary Figure 11). Here, drift detection enables the possibility of a simple way to flag these changes to users of these clinical AI models.

At the same time, there are limitations to our study. The results of this study have only been demonstrated using computer vision algorithms on chest radiographs. It is important that our findings and approach are evaluated on other types of data (e.g., tabular data, text data, or other imaging types). Likewise, the synthetic drift in this work was limited to shifts in the distribution of common attributes in patient populations. Other patient attributes such as ethnicity and race were not explored as this data is not collected at Trillium Health Partners. Those evaluating drift detection algorithms should incorporate their

context of deployment when testing attributes (e.g., previous work has shown ML models performing differently for different types of insurance[41], though this does not apply to our dataset).

Furthermore, our work only tested a small selection of all possible metrics, with a strong focus on discriminative performance metrics—the norm in most similar works evaluating and monitoring the performance of ML classifiers. Other metrics such calibration methods or clinical usefulness[42] may be more useful to detecting data-drift, as reported in past work[43]. However, our work found that the Brier score was not substantially better at detecting data drift when compared to traditional discriminative metrics. This could be because the underlying predictive model, while highly discriminative, is not well calibrated on the underlying dataset—something expected with deep learning classifiers[44,45]. It may be that improving the underlying calibration of the model may make calibration methods more useful in detecting data drift.

When implementing our experiments, we followed state-of-the-art methodologies found in previous work[46]. However, there exists a wide variety of other methods that can also be leveraged for drift detection[27] and both alternate dimensionality reduction techniques (e.g., PCA) and two-sample statistical tests (e.g., Kolmogorov–Smirnov (KS) Test or Pearson's chi-squared test depending on the data), which were not studied here.

In addition, there is a need to develop explainable or interpretable drift detection algorithms. There is currently no trustworthy automated algorithmic way of understanding what is causing the

underlying drift which makes it difficult to verify the presence of data drift or act once a drift is verified. It is here that recent work which claims to explain underlying causes of data drift using post hoc analysis (e.g., SHAP values[11,20]) can be useful. However, users of these methods should be aware of the growing body of statistical work contesting the use of SHAP values for feature explanations[24–26].

Lastly, our work uncovered a hint of inequity in drift detection: the performance of detection algorithms is not equal between different patient demographics. For example, the image-and-output-based drift detection was more sensitive to an increase in younger patients than in older patients (or male vs female). Differences between different groups of demographics was observed for all drift detection approaches. This could be a cause for concern if deployed as such methods cannot be trusted to detect drifts affecting singular groups of patients, possibly in a biased manner.

Those planning to perform drift detection have context-specific questions that they must contend with. For example, how should one determine the appropriate number of images for the target dataset when performing drift detection? While we observed that using more images increases sensitivity, this may not be possible at smaller institutions or with AI applications that see relatively few cases per day (e.g., cardiac MRIs). Institutions may be forced to trade-off between frequency of drift detection and sensitivity of drift detection. This decision should be made by consulting all relevant stakeholders while taking the full context of the deployment into account by considering the criticality of the application (i.e., how impactful is this model on patient care), the frequency of application, the population to which AI is applied (e.g., AI for a relatively static sub-population vs AI applied to a more general population), seasonal effects, among other considerations. More research is required to confidently state how data drift evaluation is performed, something that is likely also use-case specific. In addition, there should be institutional guidance on the appropriate steps to take once a drift has been detected. As more AI models are deployed in clinical care, we are hopeful that future research and development will provide more concrete guidance on how to perform data drift detection.

In conclusion, our empirical experiments on real-world healthcare data to evaluate the ability of three drift detection methods to detect data drift caused (a) naturally (emergence of COVID-19 in X-rays) and (b) synthetically, found that: (1) monitoring model performance is not a good proxy for detecting data drift, (2) data drift detection sensitivity is correlated with source and target dataset sizes, and (3) the sensitivity of data drift detection depends greatly on the specific feature being enriched. Data drift monitoring is a critical tool for maintaining reliable performance of deployed AI models to minimize risk to patient safety.

## Methods

### Dataset

Our experiments use a dataset of 239,235 chest radiographs (CXR) and associated examination reports and meta-data gathered from 78,542 patients over the age of 18 who received imaging at Trillium Health Partners, a hospital system based in Mississauga, Ontario, Canada between January 2016 and December 2020. To establish ground truth labels for each imaging study, we extracted 14 pathology labels from the associated report for each image using the CheXpert NLP algorithm[31], an approach we validated in previous work on our dataset[6]. To enable the synthetic categorical drifts, we used the associated meta-data including demographic information such as patient sex, age, hospital location, and patient type. A descriptive table of the dataset can be found in Supplementary Table 1. Supplementary Table 4 describes the performance of this model on our dataset.

**Data preprocessing.** Our original CXR dataset included 527,887 images. We excluded images that had incomplete metadata, incomplete radiological reports, or images not from a frontal view (i.e., not anterior-posterior (AP) or posterior-anterior (PA)). In addition, when more than one frontal image was present for a single study ID, we included only the last image. Lastly, to standardize the dataset, we only included images that were taken by the most commonly used imaging devices (limited to the top 10 imaging devices accounting for ~98% of images taken). This processing resulted in 239,235 frontal CXR images.

To facilitate the experiments, we further pre-processed some of the features in the metadata. We converted patients' numerical age values into three age groups of young: 18–35, middle-aged: 35–65, and senior: 65+. Lastly, we normalized and mapped some of the text-based features such as institution names (e.g., [Hospital Name] and [Hospital Name shorthand] are both mapped to [Hospital Name]).

### Data drifts

**Problem statement.** The task of drift detection is to determine whether two different sets of data (source dataset and target dataset) are from the same distribution or not. There are many different types of changes (i.e., data drifts) that can occur. In this work, we use the term data drift loosely to encompass any type of drift that may occur (e.g., covariate shift, label shift, and concept drift)[16]. For our experiments, the source dataset refers to data sampled from the original context/ setting of the algorithm. The target dataset is data sampled from the same context/setting after some time has passed. We use 4000 images to build the source and target datasets for all experiments unless otherwise specified (the sample size experiments described below). The goal is to automatically detect if enough change has occurred in the elapsed time such that the developer, researcher, or hospital administrator should take some action (e.g., retraining or taking the model offline).

In this work, we explore two types of data drift: Natural data drift (the emergence of the COVID-19) and synthetic categorical drifts. In the COVID-19 drift, the source dataset would be images taken before the COVID-19 pandemic, and the target dataset would be images taken during/after the pandemic. To test the robustness of the drift detection methods, we test the performance of detection methods on synthetic categorical drifts where we simulate changes in patient demographics. We also explore the effect of sample size on the sensitivity of drift detection approaches.

**Natural data drift (COVID-19).** The introduction of COVID-19 in 2020 presents a natural drift for our experiment. The introduction of the novel pathology produced multiple natural changes in data:

(i) the typical pattern of findings on CXR (bilateral ground glass opacities/consolidation) was uncommon before the pandemic and rapidly increased in prevalence beginning in March 2020.

(ii) because of public health policy guidelines during the pandemic in [Institution Location][47], patient encounters for non-COVID-related diseases decreased[48]. Both of these changes were obvious to human radiologists.

To facilitate the experiments, we implement a rolling window scheme where the source and target distributions are constructed by sampling 500 images from two discrete buckets (each is defined as 30 days worth of images) with 30 days between as buffer, Fig. 5. To address low numbers of patients due to public health policy at the start of the pandemic, when required to ensure each bucket has 500 images, we increase the window size of the buckets until there are enough samples for that step of the experiment. This simulates the deployment scenario where we "wait" until there are enough samples to perform the experiment within the parameters specified.

**Synthetic categorical drift.** While the COVID-19 drift represents a real-world example of a drastic change in the patient population receiving imaging, similar changes in the underlying data distribution could occur due to a variety of different factors (e.g., immigration, aging

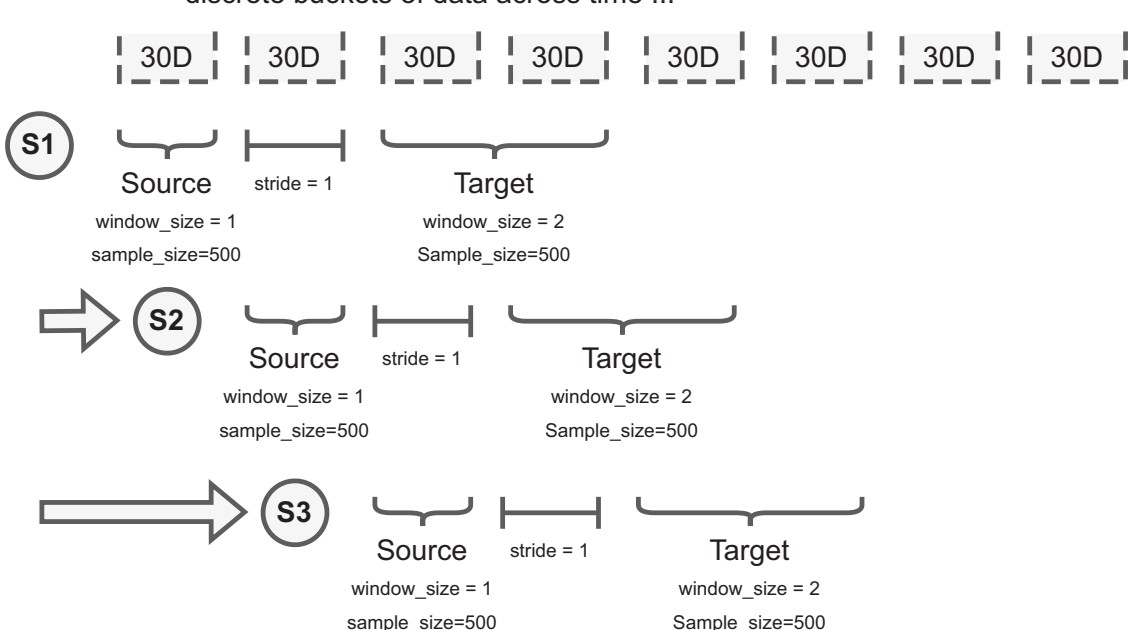

**Fig. 5 | Illustration of sampling methodology and key parameters for COVID drift detection.** We split acquired data into 30-day buckets. For example, S1, S2, and S3 represent different samples taken at different timepoints to explore the presence of data drift. For each sample we have two sub-samples: the source dataset and target dataset. Each dataset has a defined window_size (how many 30-day buckets can be used to draw patients), and sample_size (how many patients to draw from the selected window_size). Note, neither the window_size nor the sample_size of the source and target datasets need to match; the figure illustrates a difference in window_size. To increase the stability of comparisons, and avoid patients occurring in both source and target data, the stride parameter describes how many 30-day buckets separate the source and target datasets. Data are presented as mean values +/− 1 standard deviation.

population, etc.) and in a less drastic manner. As such, we examine how our drift detection works in other scenarios to determine if our findings can be generalized. To this end, we simulated the creation of synthetic drifts by sampling from our dataset and changing the demographics of the target dataset (e.g., increasing the proportion of young patients).

For the synthetic categorical drift experiments, we first define the source dataset as a stratified random sample of the overall patient population. For our experiments, we chose to stratify across the following representative categories: *sex, institution (which of the two hospitals part of [Institution Name] the images were taken), is_icu (whether the patient was admitted to the ICU), patient age,* and *patient class (whether the patient was an outpatient or inpatient).*

The distribution of the source dataset never changes. We then synthetically enrich the target dataset, a dataset with the same sample size as the source dataset, which includes two subsets of data sampled differently. The first subset is sampled using the same methodology for the source dataset. The second subset is composed of samples belonging only to the specific category being studied (e.g., male sex). For example, if our source dataset is 500 images and we are looking to explore the effect of a 10% increase in male patients, for the target dataset, we would sample 450 patients (90%) in the same manner as the source dataset for the first position of the data and add 50 male patients (10%) randomly sampled from the rest of the dataset to form the second subset of the data.

By never changing the sampling procedure of the source dataset, and slowly increasing the %of enriched samples in the target dataset, we are able to test the sensitivity of data drift approaches (described below). In the target dataset, we start with a 5% drift and keep increasing the percentage of samples iteratively (up to 50%), as illustrated in Supplementary Figure 13. In each iteration, we repeat this sampling procedure 10 times to calculate confidence intervals. To ensure the validity of the statistical test and avoid multiple tests, each dataset is composed of unique images for each statistical test.

Due to the correlation between categorical features in the data, we also monitor the increase of categorical variables that are not the intended target for the distribution shift to ensure we comprehensively catalog the changes caused in the synthetic drift. For instance, if the target category is positive instances of Pneumonia, this may also correlate with other pathologies, such as Lung Opacity, that are symptomatic of the target category[49]; this is to be expected and unavoidable due to the limitations of the size and diversity of the dataset. The proportion of the increase in non-target categories for each experiment that rises above a tolerance of 5% from the proportions in the full source dataset is included in the supplemental material (Supplementary Tables 2 and 3).

### Automated drift detection methodology

With the experimental setup defined, we test the ability of model performance monitoring to detect data drift in addition to three drift detection approaches: (1) image data-based drift detection, (2) model output-based drift detection, and (3) combined image-and-output-based drift detection. The CyclOps package was used to conducted the experiments[50].

**Tracking model performance (alternatively: model performance-based drift detection).** Tracking model performance (i.e., performance-based drift detection) is commonly used to detect if input distributions have shifted. Intuitively, if the performance of the model falls below (or exceeds) expected norms, it is suspected that the inputs to the model (i.e., the data) has changed. In this work, we monitor the performance of a TorchXRayVision model (described below) fine-tuned on a small subset of patient X-ray images ($n = 10,000$, which were not used in the rest of the work). We measure performance using AUROC, F1, precision (alternatively: positive predictive value), and recall (alternatively: sensitivity) because these are the most commonly reported metrics in the relevant literature (i.e., past work on chest X-ray classification and data drift detection). We

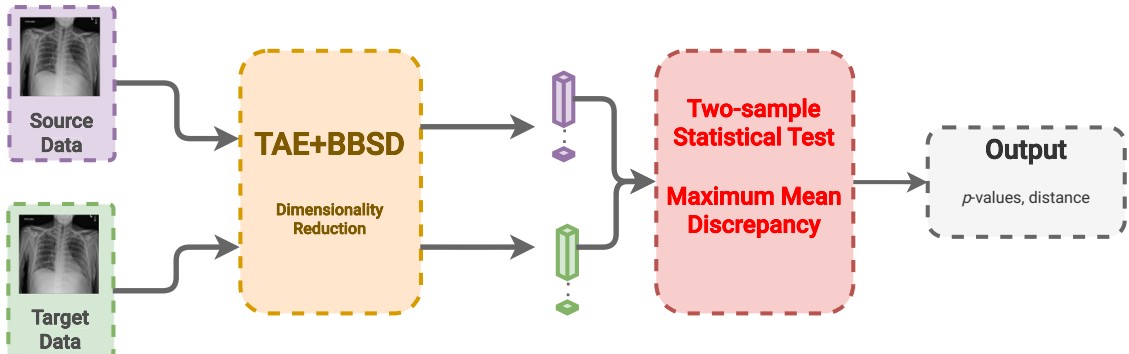

**Fig. 6 | The flowchart for the TAE + BBSD method.** CXR images from the source and target sets are fed into both a trained autoencoder (TAE) and a black box shift detector (BBSD) to create a feature vector of each image with reduced dimensionality. The feature vector from the bottleneck of the TAE and the softmax probabilities from the classifier are concatenated and passed as one feature vector to the two-sample statistical test to determine if drift has occurred. Data are presented as mean values +/− 1 standard deviation.

also include the Brier Score Loss (i.e., Brier Score) in our experiments as a measure of calibration metrics.

In cases where there are not enough predicted positive or negative instances for a given pathology (machine learning task described below), we exclude those classes while measuring the model performance.

**Image data-based drift detection.** Image data-based drift detection (also: feature/covariate shift detection) describes an approach to detect drift by comparing the set of images in the source and target datasets directly. To do this, the patient chest radiographs are passed into a convolution neural network autoencoder to build image representations (i.e., reduce the dimensionality to a feature vector with 512 dimensions). In this work, we use a pre-trained TorchXRayVision AutoEncoder (TAE)[51]. After building the representation of each image, we compare the reference and target datasets using the multivariate maximum mean discrepancy (MMD) statistical test[46,52].

MMD is a nonparametric statistical technique for detecting distributional differences between two samples[53]. As demonstrated by Rabanser et al.[46], the p-values are calculated by computing the maximum discrepancy between two distributions and comparing via permutation tests to a null-hypothesis distribution.

**Model output-based drift detection (alternatively: blackbox shift detection; BBSD).** Model output-based drift detection is a method to detect drift using only the output of the classifier without any gold labels[46]. Rather than reducing the dimensionality of the images using an auto-encoder, this approach reduces the dimensionality of the images through the act of classification into a vector with a length equal to the number of predicted classes (14 in this case). To clarify related concepts, data drift using BBSD uses the model predictions as a proxy for data changes, while target drift (similar concept) would describe changes in the ground truth outcomes (e.g., decrease in bilateral pneumonias from introduction of the COVID-19 vaccine). Likewise, BBSD is different from performance-based drift detection as we do not require any gold labels for this method, only the predicted outputs. Like with TAE, after dimensionality reduction we compare the reference and target datasets using the MMD statistical test[46,52].

**Combined image-and-output-based drift detection (TAE + BBSD).** In this last approach, we combine the previous two approaches using the images (TAE) in conjunction with the model outputs (BBSD). For the images, we reduce their dimensionality using the aforementioned TAE. The classifier is used as a black-box shift detector (BBSD)[54], hence TAE + BBSD. These two outputs (from the autoencoder and the classifier) are normalized independently, concatenated and used as the final feature vector for performing the statistical test with the chosen test method, MMD, Fig. 6. Comparing the performance between TAE, BBSD, and TAE + BBSD enables us to observe if the type of drift captured by the auto-encoder differs from the drift captured by the classifier.

### Models and training
**TorchXRayVision classifier.** The classifier used for the performance monitoring and BBSD is the classifier from the TorchXRayVision library[51]. The pre-trained version of the TorchXRayVision used in this study was pre-trained on the MIMIC-CXR dataset[55]. The model architecture is a Densenet-121 convolutional neural network that takes an input of 224 × 224 images and outputs a set of 14 predictions.

We further trained and fine-tuned the model on [Institution Name]'s training set and used a validation dataset for early stopping and hyperparameter tuning. The training set was composed of 10,000 scans of unique patients from January 2016 to June 2016 and validated on data from July 2016 to December 2016. There was no patient overlap between the source and target sets for both the COVID and synthetic drifts experiments. A table with model performance metrics can be found in the Supplemental Materials.

**TorchXRayVision autoencoder.** The pre-trained TorchXRayVision autoencoder was trained on numerous chest radiograph datasets (PadChest[56], Chestx-ray8[57], CheXpert[31], and MIMIC-CXR[55]). The autoencoder takes in images of size 224 × 224 and uses a resnet-101 backbone to reduce the dimensionality of the image to feature map of size 4 × 4 × 512. This feature map is then fed into an inverted resnet-101 backbone, the objective is to reconstruct the input images so that the feature maps at the bottleneck of the network contain useful information about the scans. For the experiments, we take the feature map produced by the autoencoder and perform mean pooling to produce a 1 × 512 feature vector for each scan.

### Inclusion & ethics
All stages of the research were conducted by local researchers. The study was deemed locally relevant by the Institute for Better Health and the protocol was approved by the research ethics board (#1031).

### Reporting summary
Further information on research design is available in the Nature Portfolio Reporting Summary linked to this article.

## Data availability
The dataset from this study is held securely in coded form at the Institute for Better Health, Trillium Health Partners and is not openly

available due to privacy concerns. However, access may be granted to those who meet criteria for confidential access, please contact Mohamed Abdalla or Benjamin Fine (at first.last@thp.ca).

## Code availability

All code used to create all experiments in this study can be found publicly available at https://doi.org/10.5281/zenodo.10652201

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

## Acknowledgements

We would like to thank the TD Ready Commitment and Digital Supercluster Canada which provided financial support to B.F. Neither had say in the decision to pursue or the decision to publish this research.

## Author contributions

A.K.: Data curation, formal analysis, methodology, software, visualization, writing—original draft. E.B.: Data curation, formal analysis, methodology, software. V.S.: Methodology, writing—review & editing. Mou.A.: Methodology, writing—review & editing. B.F.: Conceptualization, funding acquisition. E.D.: Conceptualization, methodology. Moh.A.: Conceptualization, data curation, formal analysis, methodology, writing—original draft.

## Competing interests

B.F.: Support from Trillium Health Partners Foundation and Digital Supercluster Canada; leadership role on HaloHealth Angel Network Board; stocks or stock options in PocketHealth, Phelix, and Eva Medical. None of these interests had a say in pursuing or publishing this research. The remaining authors have no competing interests to declare.
