## [Peer Review File · Nature Communications]

Empirical data drift detection experiments on real-world medical imaging dataREVIEWER COMMENTS

Reviewer #1 (Remarks to the Author):

The authors developed several methods to detect data drift: image data-based drift detection (TAE), model output-based drift detection (BBSD), combined image-output (TAE+BBSD). They performed a sensitivity analysis on real-world data consisting of chest X-rays prior to and during the COVID-19 pandemic and also by using sampling techniques to modify the percentage of certain categorical variables associated with the patients such as age range and gender, etc. They demonstrated that detecting changes in discriminative performance metrics like AUC, precision, recall, and F1 aren't sufficient for determining whether a model should be updated. The results from this work are consistent with what would be expected, and the sensitivity analysis was thorough, but there is a major concern that some fundamental concepts, namely calibration, are missing and should be integrated into this manuscript. A relevant reference is chapter 19 from the book "Clinical Prediction Models Second Edition" by Ewout W. Steyerberg.

1. The performance metrics (AUC, precision, recall, F1) are strictly discriminative performance metrics. Other categories include calibration and clinical usefulness (Steyerberg section 19.2.2 page 371). The data drift, or changes in case mix, that is detected in this study is important because, even though it doesn't have much of an effect on these discriminative performance metrics, it will have an effect on calibration performance metrics such as Harrell's E statistics (e.g. Eavg, Emax, etc.), the Brier score, or the integrated calibration index. When models are no longer calibrated, they need to be updated or retrained. The concept of calibration performance drift should be introduced and integrated. This study utilizes neural networks, but the Steyerberg book and several other references discuss calibration in terms of generalized linear models (GLMs). This study also applies drift detection methods that don't require labels and methods based on discriminative or calibration performance metrics do require labels.

2. Other studies have also shown that AUC can remain unchanged in the presence of data/calibration drift. One such example is a study on a COVID-19 predictive model (Levy et al. "Development and validation of self-monitoring auto-updating prognostic models of survival for hospitalized COVID-19 patients" Nature Communications 2022), and there are several other papers in the literature on the detection of calibration drift as it relates to model updating. The authors need to address whether it would be expected that detecting data drift without labels will be advantageous with respect to other procedures that monitor calibration performance drift and showcase the superiority of this approach (or equivalency, since it does offer a slight advantage of not needing the label).

Other comments are as follows:

3. This paper empirically performs a power analysis to determine the minimum number of images for the target dataset. This is appropriate, but the explanations of these observations are lacking. For instance, it was stated that it was not clear why a 5% increase in 18-35 year old patients was detected but a 30% increase in 65+ patients was required for detection and that inpatients were better detected than outpatients. Only one hypothesis was discussed; that of the minority class. Could this be related to the amount of discrimination between these cohorts with larger covariances and smaller separation of means leading to a larger number of samples required? There are also references such as Riley et al. "Calculating the sample size required for developing a clinical prediction model" that may give some insight into which aspects of the data may influence the required number of samples but that is also framed in terms of GLMs.

4. The second paragraph in the discussion provides an example where timely evaluation is when outcomes occur within days or weeks. This is application dependent. The COVID-19 example exhibits very sudden changes, but many other applications that are routinely updated such as predicting outcomes from heart surgery (e.g. Euroscore) or the onset of cardiovascular disease (e.g. QRisk) may require updates on the timescale of years or decades. Just make sure that the wording of this example statement generalizes.

5. Some typos:

a. The Figure 4 caption is missing an s and should state "The plots on the right" because there are two plots.

b. The fifth paragraph in the discussion: "there is" no change in performance in detecting ...

c. The fifth paragraph in the discussion: "events" is in the wrong location. As such, pharmaceutical companies are being increasingly asked to actively monitor adverse across broad demographics

events after approval and sales to patients

d. Table 1 Scenario c: hideany needs a space

e. The fourth numbered item in the automated drift detection methodology section has an extra closing parenthesis after TAE+BBSD.

6. The limitations discuss evaluating this method on tabular data, and Steyerberg or a related paper could be cited here.

Reviewer #2 (Remarks to the Author):

Overall this is an article with a clear and concise point: make sure you look at data going into your model to assess data drift, not just what the model is doing with it. This is useful since looking at model performance alone is 1) delayed (since you have to collect ground truth over time) and 2) un-insightful since data drift can occur without much change to aggregated performance metrics. The authors present these findings with a couple of data drift experiments evaluating input and output (and both) vs performance.

Despite some interesting results, I have some major concerns about the paper in its current format that need to be addressed before considering publication:

1) At points I find the description of data drift confused. This is, in part, due to the headaches of terminology in this field but the authors tend to refer to data drift to a number of things that could change in a system with a deployed model. I think what would help here is reference to specific types of drift (i.e., covariate shift, prior probability shift, concept drift) to help motivate their approach.

For example, in the Discussion the authors argue that "those using clinical AI should also monitor for data drift". This is too generic. I assume they are arguing for using TAE+BBSD (i.e., monitoring covariate shifts and prior probability shifts) along with performance metrics. Then explain explicitly how this would help (e.g., TAE helps monitor covariate shift, BBSD helps monitor prior probability shift and combining this with performance metrics we could even try to monitor if concept drift is occurring - the latter point being important since we are looking for concept drift really, not just the covariates drifting to an area of lower local performance in the parameter space).

Despite this, I appreciate the point of the article is to "look at the data rather than the model". Ok but what happens if the distributions of input and output remain the same and the concept changes? In reality, these measures are complimentary not either or as the article is (sometimes) positioned.

On this note, I find direct comparison with performance metrics (Figures 3-8) to track drift confusing, considering you suggest combining use with TAE+BBSD in the discussion. They are measuring different things (in reality performance tracking is a poor & low effort way of tracking concept drift - or at least a basic notion of when to re-train), whereas you are looking at covariate shift here. I'm not sure anyone is trying to claim tracking performance is a good way of specifically tracking covariate shift so I find the statements a bit odd. It is probably better to frame it with the added value it brings (as you do in the discussion).

2) In Figure 2 you demonstrate how performance, TAE etc. change with increasing shifts in covariates. Are these covariates actually important for prediction? They appear to be secondary to more predictive features in the image, so why would we expect there to be changes in the performance?

3) The manuscript contains multiple criticisms of other data drift detection approaches despite appearing to miss the point of why they are used (and actually that they could be complimentary). Overall the article is quite combative.

For example the authors criticise post-hoc explanations without stating why they are actually

considered. Explanations are useful since they can also provide immediate insight (you don't need gold labels), provide relevant context about the changing situation (as 'perceived' by the model) locally (i.e., per prediction) and provide context on whether it is a drift you should be concerned about (i.e., if the most predictive features are changing).

The real criticism is that they are still model-centric and don't explicitly point to covariate drift (which you do start to allude to). Despite this, there is obviously utility in looking at how this impacts the workings of the model as well. I do wonder if this more complimentary to TAE-BBSD than either-or as you seem to state. Even if you have lost faith with the field of additive explanations, there are other methods to consider (e.g., counterfactuals).

This is similar to point 1) where you compare performance metrics against TAE+BBSD. Yes this is better practice but are they not looking at different things and better phrased as complimentary?

We would like to thank the editorial team and reviewers for their efforts and thoughtful comments. Below, we respond point by point to the reviewer comments. The original text is indented and in black, our response is in blue.

REVIEWER COMMENTS

Reviewer #1 (Remarks to the Author):

The authors developed several methods to detect data drift: image data-based drift detection (TAE), model output-based drift detection (BBSD), combined image-output (TAE+BBSD). They performed a sensitivity analysis on real-world data consisting of chest X-rays prior to and during the COVID-19 pandemic and also by using sampling techniques to modify the percentage of certain categorical variables associated with the patients such as age range and gender, etc. They demonstrated that detecting changes in discriminative performance metrics like AUC, precision, recall, and F1 aren't sufficient for determining whether a model should be updated. The results from this work are consistent with what would be expected, and the sensitivity analysis was thorough, but there is a major concern that some fundamental concepts, namely calibration, are missing and should be integrated into this manuscript. A relevant reference is chapter 19 from the book "Clinical Prediction Models Second Edition" by Ewout W. Steyerberg.

We thank the reviewer for their insightful review. We agree that that the analysis conducted thus far focused only on discriminative performance metrics. Below, in response to point 1, we present new experiments performed to evaluate the effect of calibration metrics.

1. The performance metrics (AUC, precision, recall, F1) are strictly discriminative performance metrics. Other categories include calibration and clinical usefulness (Steyerberg section 19.2.2 page 371). The data drift, or changes in case mix, that is detected in this study is important because, even though it doesn't have much of an effect on these discriminative performance metrics, it will have an effect on calibration performance metrics such as Harrell's E statistics (e.g. Eavg, Emax, etc.), the Brier score, or the integrated calibration index. When models are no longer calibrated, they need to be updated or retrained. The concept of calibration performance drift should be introduced and integrated. This study utilizes neural networks, but the Steyerberg book and several other references discusses calibration in terms of generalized linear models (GLMs). This study also applies drift detection methods that don't require labels and methods based on discriminative or calibration performance metrics do require labels.

We agree that that the analysis conducted thus far focused only on discriminative performance metrics. To address this, we have redone all experiments in the paper having added a new calibration metric (Brier Score). This has resulted in changes to the results, discussion, and methods sections as well as the updating of many figures in the Appendix (new Figure 9 and 10, and updated Figure 12).

While we found that the Brier Score can be more sensitive than other discriminative metrics, this increase in sensitivity is not consistent nor particularly substantial in our dataset. We hypothesize that this may be due to a lack of calibration of the underlying model – deep neural networks especially those convolutional based have been shown repeatedly to have low calibration (something which we have

now added to the discussion of the paper). We have also pointed the reader to the reference above and highlighted how our findings may change with the consideration of other metrics.

2. Other studies have also shown that AUC can remain unchanged in the presence of data/calibration drift. One such example is a study on a COVID-19 predictive model (Levy et al. “Development and validation of self-monitoring auto-updating prognostic models of survival for hospitalized COVID-19 patients” Nature Communications 2022), and there are several other papers in the literature on the detection of calibration drift as it relates to model updating. The authors need to address whether it would be expected that detecting data drift without labels will be advantageous with respect to other procedures that monitor calibration performance drift and showcase the superiority of this approach (or equivalency, since it does offer a slight advantage of not needing the label).

This comment relates closely with the comments raised by the second reviewer. We have updated the discussion of our paper to lessen the confrontational nature of our paper – it was not our intention to convince readers that label-based analyses had no worth. Rather, that they were not sufficient to ensuring safety in all situations. We have updated the discussion accordingly to discuss the benefit and utility of both approaches and incorporated the measuring of calibration as one of these approaches for detecting data drift.

Other comments are as follows:

3. This paper empirically performs a power analysis to determine the minimum number of images for the target dataset. This is appropriate, but the explanations of these observations are lacking. For instance, it was stated that it was not clear why a 5% increase in 18-35 year old patients was detected but a 30% increase in 65+ patients was required for detection and that inpatients were better detected than outpatients. Only one hypothesis was discussed; that of the minority class. Could this be related to the amount of discrimination between these cohorts with larger covariances and smaller separation of means leading to a larger number of samples required? There are also references such as Riley et al. “Calculating the sample size required for developing a clinical prediction model” that may give some insight into which aspects of the data may influence the required number of samples but that is also framed in terms of GLMs.

We thank the reviewer for this additional hypothesis, one which we believe is plausible and worth testing. We have included this additional hypothesis and citation following our original hypothesis.

4. The second paragraph in the discussion provides an example where timely evaluation is when outcomes occur within days or weeks. This is application dependent. The COVID-19 example exhibits very sudden changes, but many other applications that are routinely updated such as predicting outcomes from heart surgery (e.g. Euroscore) or the onset of cardiovascular disease (e.g. QRisk) may require updates on the timescale of years or decades. Just make sure that the wording of this example statement generalizes.

We agree that the approach described in our paper is not needed and may not be the best for all circumstances. We have further emphasized this in the relevant paragraph in the discussion.

5. Some typos:

- a. The Figure 4 caption is missing an s and should state “The plots on the right” because there are two plots.
- b. The fifth paragraph in the discussion: “there is” no change in performance in detecting ...
- c. The fifth paragraph in the discussion: “events” is in the wrong location. As such, pharmaceutical companies are being increasingly asked to actively monitor adverse across broad demographics events after approval and sales to patients
- d. Table 1 Scenario c: hide any needs a space
- e. The fourth numbered item in the automated drift detection methodology section has an extra closing parenthesis after TAE+BBSD.

We have corrected all of the typos listed above.

6. The limitations discuss evaluating this method on tabular data, and Steyerberg or a related paper could be cited here.

In the discussion we have stated that our findings and results may not hold for different types of data (explicitly mentioning tabular data).

Reviewer #2 (Remarks to the Author):

Overall this is an article with a clear and concise point: make sure you look at data going into your model to assess data drift, not just what the model is doing with it. This is useful since looking at model performance alone is 1) delayed (since you have to collect ground truth over time) and 2) un-insightful since data drift can occur without much change to aggregated performance metrics. The authors present these findings with a couple of data drift experiments evaluating input and output (and both) vs performance.

Despite some interesting results, I have some major concerns about the paper in its current format that need to be addressed before considering publication:

1) At points I find the description of data drift confused. This is, in part, due to the headaches of terminology in this field but the authors tend to refer to data drift to a number of things that could change in a system with a deployed model. I think what would help here is reference to specific types of drift (i.e., covariate shift, prior probability shift, concept drift) to help motivate their approach.

For example, in the Discussion the authors argue that "those using clinical AI should also monitor for data drift". This is too generic. I assume they are arguing for using TAE+BBSD (i.e., monitoring covariate shifts and prior probability shifts) along with performance metrics. Then

explain explicitly how this would help (e.g., TAE helps monitor covariate shift, BBSD helps monitor prior probability shift and combining this with performance metrics we could even try to monitor if concept drift is occurring - the latter point being important since we are looking for concept drift really, not just the covariates drifting to an area of lower local performance in the parameter space).

We thank the reviewer for this thoughtful discussion of our work and agree with the points raised. To address the confusion regarding data drift, we have reworked the first couple of paragraphs in the introduction to explicit add a definition and a citation for the definition. We also add relevant points raised by the reviewers (how different approaches enable detection of different possible changes to the data).

Despite this, I appreciate the point of the article is to "look at the data rather than the model". Ok but what happens if the distributions of input and output remain the same and the concept changes? In reality, these measures are complimentary not either or as the article is (sometimes) positioned.

We agree that, in an attempt to convince the reader of the importance of our findings and approach, our initial phrasing was too "either or", and we have updated the body of the text to highlight that our approach complements existing approaches and is not meant to replace them. These changes have taken place largely in the introduction and discussion sections with minor changes in the title and abstract.

On this note, I find direct comparison with performance metrics (Figures 3-8) to track drift confusing, considering you suggest combining use with TAE+BBSD in the discussion. They are measuring different things (in reality performance tracking is a poor & low effort way of tracking concept drift - or at least a basic notion of when to re-train), whereas you are looking at covariate shift here. I'm not sure anyone is trying to claim tracking performance is a good way of specifically tracking covariate shift so I find the statements a bit odd. It is probably better to frame it with the added value it brings (as you do in the discussion).

We agree with the reviewer that they are measuring different things, however there are works that justify/validate the performance of their approaches using discriminative metrics (such as: <https://arxiv.org/pdf/2202.02833.pdf>). As such, we have kept the figures, but have added a few sentences emphasizing the points raised by the reviewer: that they are measuring different things and as such, tracking model performance should not be expected to detect model drift in all instances.

2) In Figure 2 you demonstrate how performance, TAE etc. change with increasing shifts in covariates. Are these covariates actually important for prediction? They appear to be secondary to more predictive features in the image, so why would we expect there to be changes in the

performance?

Similar to the above point, it's not that we hypothesized that the tested features would be important for prediction/would affect prediction. We just wanted to comprehensively evaluate the effect of changing the dataset and the effect that had on performance (in response the work cited above).

3) The manuscript contains multiple criticisms of other data drift detection approaches despite appearing to miss the point of why they are used (and actually that they could be complimentary). Overall the article is quite combative.

For example the authors criticise post-hoc explanations without stating why they are actually considered. Explanations are useful since they can also provide immediate insight (you don't need gold labels), provide relevant context about the changing situation (as 'perceived' by the model) locally (i.e., per prediction) and provide context on whether it is a drift you should be concerned about (i.e., if the most predictive features are changing).

The real criticism is that they are still model-centric and don't explicitly point to covariate drift (which you do start to allude to). Despite this, there is obviously utility in looking at how this impacts the workings of the model as well. I do wonder if this more complimentary to TAE-BBSD than either-or as you seem to state. Even if you have lost faith with the field of additive explanations, there are other methods to consider (e.g., counterfactuals).

This is similar to point 1) where you compare performance metrics against TAE+BBSD. Yes this is better practice but are they not looking at different things and better phrased as complimentary?

We agree with the reviewer that in our attempt to prove the utility of our described methods, we may have not perfectly phrased the utility of other approaches. We have updated the title, a sentence in the abstract to highlight the complementary nature of both approaches, and made more substantive changes to the introduction and discussion sections of the paper to better reflect the situation as the reviewer describes.

REVIEWER COMMENTS

Reviewer #1 (Remarks to the Author):

Thank you for addressing my comments and concerns and for incorporating the Brier score calibration metric into your manuscript. I would have expected that the calibration performance would have degraded as the data changed, but since this wasn't what you found, you still need to demonstrate what problem you're trying to solve. Your introduction motivates this work by warning that data drifts can hypothetically result in algorithm underperformance and, as a result, harm to the patient, but your results show changes in the data without corresponding changes in discriminative or calibration performance. It's important to be able to show why these data changes matter in your real and simulated datasets. Otherwise detecting data drift is a solution to a merely hypothetical problem. One place to show this would be in Figure 1 where you show that the discrimination performance is nearly constant while the data drift is occurring. If both discriminative or calibration performance doesn't change in this figure, maybe metrics from net-benefit curves, or other performance metrics that would be clinically relevant might show significant changes. In general, I believe it is imperative to back the claim that the detected data drift results in some model performance issue. Absent of that, the importance of data drift detection needs to be motivated not by a hypothetical (and based on the data showed, non-existent) performance drift, but for other reasons, that need to be explained, both in the introduction and discussion of this manuscript. I believe this is a major point that needs to be resolved, to establish the significance of this work - if it is a theoretical exercise, this paper is not of wide interest and would be better suited in a purely medical informatics methodological journal.

Also, please note the typo on page 4 "rather; rather". This sentence actually mentions that [discriminative] model performance isn't necessarily a good predictor of data drift. But my main concern is that data drift only matters because it can affect model performance, and you need to show why we should care about data drift.

I also noticed that you added new references to the end of the reference list, and that these references are not in the order in which they appear in the manuscript.

Reviewer #2 (Remarks to the Author):

As per my first review: this is an article with a clear message - i.e., to understand the continued validity of a deployed ML model you need to understand the data going in (not just the model performance or explanations).

I'm happy the authors have addressed my concerns that it is not either/or and that the combination of multiple techniques (and hence the more info you have to hand), the better.

We would like to thank the editorial team and reviewers for their efforts and thoughtful comments on our revision. Below, we respond point by point to the reviewer comments. The original text is indented and in black, our response is in blue.

REVIEWER COMMENTS

Reviewer #1 (Remarks to the Author):

Thank you for addressing my comments and concerns and for incorporating the Brier score calibration metric into your manuscript. I would have expected that the calibration performance would have degraded as the data changed, but since this wasn't what you found, you still need to demonstrate what problem you're trying to solve. Your introduction motivates this work by warning that data drifts can hypothetically result in algorithm underperformance and, as a result, harm to the patient, but your results show changes in the data without corresponding changes in discriminative or calibration performance. It's important to be able to show why these data changes matter in your real and simulated datasets. Otherwise detecting data drift is a solution to a merely hypothetical problem. One place to show this would be in Figure 1 where you show that the discrimination performance is nearly constant while the data drift is occurring. If both discriminative or calibration performance doesn't change in this figure, maybe metrics from net-benefit curves, or other performance metrics that would be clinically relevant might show significant changes. In general, I believe it is imperative to back the claim that the detected data drift results in some model performance issue. Absent of that, the importance of data drift detection needs to be motivated not by a hypothetical (and based on the data showed, non-existent) performance drift, but for other reasons, that need to be explained, both in the introduction and discussion of this manuscript. I believe this is a major point that needs to be resolved, to establish the significance of this work - if it is a theoretical exercise, this paper is not of wide interest and would be better suited in a purely medical informatics methodological journal.

As the paragraph above includes many points, we will split it for our response.

- *Your introduction motivates this work by warning that data drifts can hypothetically result in algorithm underperformance and, as a result, harm to the patient, but your results show changes in the data without corresponding changes in discriminative or calibration performance.*

We agree with the reviewer that we could have more clearly stated the need for data drift detection lest it be considered a hypothetical problem. In the previous version, the motivation for drift detection was left mostly to the discussion. However, we have now also included this in the introduction. We summarize the points below.

First, the ability of data-drift detection methods to detect changes that directly affect model performance is well described in the literature. We now highlight multiple prior works in our introduction. However, we emphasize that this is not the main contribution of our work.

- It's important to be able to show why these data changes matter in your real and simulated datasets. Otherwise detecting data drift is a solution to a merely hypothetical problem.

We agree. Consider a case where there has been a substantial drift in the data distribution but no detectable change in performance. Is this important/does this affect patient safety? We would argue yes and can do so using 2 main directions.

1. AI-Safety principles: Technical best practice and emerging regulatory guidance (FDA, 2023) requires that an algorithm only be used after being thoroughly evaluated on a particular population. This is likely to avoid the risk of potential harm, degradation or bias, which may be unknown or unmeasured. If a population changes by 50%, technical experts and regulatory guidelines would indicate that models applied to this population should undergo thorough evaluation (more than just monitoring performance). In this way, drift-detection enables us to trigger re-evaluation to adhere to technical and regulatory best practices – thus ensuring patient safety.
2. New diseases in your populations: Consider an algorithm trained to detect only 3 diseases (Edema, Pleural Effusion, and Nodules). When a new disease becomes rapidly prevalent in the population (e.g., say a large bout of pneumonia starts spreading), it's reasonable to expect the deployed algorithm to keep working for the three trained diseases. However, care providers would likely prefer to be warned that:
 - a. There is a substantial change in your population (which could be a new disease). Knowing about this can improve patient safety/outcomes.
 - b. Depending on the use case, e.g., if the algorithm was used for triaging or collated into normal vs abnormal classifier, then continued use, without updating, would be inappropriate. This is another instance where performance does not decrease, but patient safety/outcomes could be better improved given

We discuss more theoretical points in the discussion of the paper, but believe the above points demonstrate how this provides practical utility to institutions today.

- One place to show this would be in Figure 1 where you show that the discrimination performance is nearly constant while the data drift is occurring.

The discriminative performance of the classifier does not change because of the second point raised in our response to the last point – this is not a class our algorithm was trained on or evaluated on. Our classifier can detect consolidation and other observations caused by COVID, but cannot assign it to a COVID class. For example, many COVID cases would have the algorithm detect atelectasis but a radiologist (who has learned the patterns of COVID) would instead label the image as having possible COVID pneumonia. This is an “error” (if the AI algorithm is meant to capture disorders or observations caused by observations) that can't be captured by automated aggregate performance measures. This is why we advocate for the use of drift detection to inform users when a more thorough evaluation is required.

We hope that our above response has highlighted how drift detection is of practical utility to practitioners. We have updated the introduction to make this clearer.

Also, please note the typo on page 4 “rather; rather”. This sentence actually mentions that [discriminative] model performance isn’t necessarily a good predictor of data drift. But my main concern is that data drift only matters because it can affect model performance, and you need to show why we should care about data drift.

We have fixed the typo. As for discussion regarding the justification for data drift, please see our response to the first point.

I also noticed that you added new references to the end of the reference list, and that these references are not in the order in which they appear in the manuscript.

We have updated the numberings of the citations. Citations should now all be in order.

Reviewer #2 (Remarks to the Author):

As per my first review: this is an article with a clear message - i.e., to understand the continued validity of a deployed ML model you need to understand the data going in (not just the model performance or explanations).

I'm happy the authors have addressed my concerns that it is not either/or and that the combination of multiple techniques (and hence the more info you have to hand), the better.

We thank the reviewer for this feedback. We're happy that our edits have successfully conveyed our agreement that multiple techniques are required.